# WHEN SELF-SUPERVISED LEARNING MEETS UNBOUNDED PSEUDO-LABEL GENERATION

## ABSTRACT

Self-supervised learning (SSL) has demonstrated strong generalization abilities across diverse downstream tasks. However, it is difficult for SSL to accurately gather samples of the same category and separate samples of different categories in the training stage. In this paper, we present a novel approach of generating pseudo-labels for augmented samples to regulate their feature-space relationships. To align the pseudo-label space with the ground-truth label space, we propose an instance-level pseudo-label generation mechanism. Building upon our observations that pseudo-labels can encompass unbounded label noise and that learning remains robust to such noise in the early stages of training, we propose Precise Adjustment Regularization (PAR) for precise dynamic relationship mining. Finally, we propose a PAR-based bi-level optimization learning mechanism mechanism (PBOLM) to promote high-quality representations in SSL. Theoretically, from a data generation perspective, we demonstrate that the proposed PBOLM is more conducive to extracting critical generative factors in data generation. Empirically, based on various downstream tasks, we demonstrate that PBOLM can be considered a plug-and-play module to enhance the performance of SSL methods.

## 1 INTRODUCTION

Unsupervised representation learning is a significant research area in machine learning. One recent breakthrough is self-supervised learning (SSL), which has demonstrated impressive performance in various computer vision tasks, including classification, object detection or segmentation, and transfer learning (Jaiswal et al., 2020; Si et al., 2022; Radford et al., 2021). A characteristic of SSL is its instance-based learning mechanism, which treats each sample as an independent class. This also enables SSL to extract semantic information directly from the data, which can then be transferred effectively to diverse downstream tasks. Unless specified otherwise, SSL in this paper refers to both contrastive and non-contrastive learning methods related to data augmentation invariance.

An essential concern in SSL is that it faces significant challenges during the training process due to the inherent absence of labeled information. This limitation has a direct impact on their capacity to accurately model the dynamic relationships that exist among augmented samples. Specifically, SSL predominantly concentrates on aggregating augmented samples originating from the same ancestral source, inadvertently neglecting the aggregation of augmented samples derived from ancestor samples that share identical labels. Furthermore, SSL tends to confine itself to imposing prior knowledge on the overall data distribution, inadvertently overlooking the crucial concept that augmented samples resulting from ancestor samples with different labels should ideally exhibit a tendency to diverge or move apart from each other.

To tackle the problem outlined above, we propose to generate pseudo-labels to guide the learning of SSL. Nonetheless, this strategy has not been easy either. Firstly, achieving precise aggregation and separation for different augmented samples requires prior knowledge of the sample label space. However, due to the unknown nature of labels during the training phase, directly acquiring labels through clustering can result in an uncontrollable and incorrect number of classes. Therefore, aligning the pseudo-label space with the true label space poses a critical challenge. Secondly, even if the first challenge is effectively addressed, the acquired pseudo-labels may still contain label noise. Training models directly with noisy labels can significantly degrade the model's ability to generalize. Consequently, robust learning strategies under label noise become a pivotal challenge. Lastly,

even if we achieve perfect robust learning based on pseudo-labels, integrating this with SSL methods remains a significant challenge, because the ultimate goal is to employ pseudo-labels to fine-tune SSL methods, enabling precise clustering of similar samples and separation of dissimilar ones.

In this paper, we first propose an instance-level pseudo-label generation mechanism that effectively controls dynamic relationships between augmented and anchor samples, mitigating label space inconsistencies. Then, we prove that pseudo-labels generated by our mechanism are unbounded, and we demonstrate that even in this setting, the learning model has an aptitude for accurately capturing dynamic changes in the early stages of training. Thus, we propose a novel approach known as Precise Adjustment Regularization (PAR), which leverages early training properties for precise dynamic relationship mining among training samples. Finally, we propose a PAR-based bi-level optimization learning mechanism (PBOLM)) that harnesses pseudo-label learning to enhance the quality of SSL representations. We explain why PBOLM can restore generalization from the Perspective of data generation. We also validate the effectiveness of PBOLM through extensive experiments.

The main contributions of this paper are as follows: (1) We propose an instance-level pseudo-label generation mechanism that ensures the generated pseudo-label space aligns with the ground-truth label space. Furthermore, we establish that these pseudo-labels contain unbounded label noise; (2) We provide evidence that learning with unbounded pseudo-labels exhibits robustness during the early stages of training. To capitalize on this insight, we propose Precise Adjustment Regularization (PAR) as a means to enhance learning with unbounded pseudo-labels; (3) We present PBOLM, a framework designed to augment SSL methods by improving their ability to aggregate similar data points and separate dissimilar ones during the training process. (4) Theoretical analysis from a data generation perspective demonstrates that PBOLM can effectively capture additional data generation factors, leading to improved generalization. We empirically validate the efficacy of PBOLM across a diverse set of downstream tasks, showcasing its practical utility.

## 2 RELATED WORK

**Learning Method.** SimCLR (Chen et al., 2020a) is the first widely used CL method that achieves comparable performance to supervised learning. However, SimCLR requires large training batches and high computational resources. MoCo (He et al., 2020; Chen et al., 2020b; 2021b) addresses this issue by using dynamic memory allocation. MetAug (Li et al., 2022) generates hard positive samples to reduce the negative sample redundancy. Some methods avoid negative samples altogether, such as BYOL (Grill et al., 2020), W-MSE Ermolov et al. (2021), Simsiam Chen & He (2021), and Barlow Twins Zbontar et al. (2021). However, these methods ignore the intrinsic structure of the data distribution. SwAV (Caron et al., 2020) and PCL (Li et al., 2020) exploit the clustering structures embedded in the data distribution. LMCL (Chen et al., 2021a) mines the large margin between positive and negative samples. CL can be seen as an instance-based learning paradigm, which limits its ability to capture the relationship between different instances. ReSSL (Zheng et al., 2021; Tomasev et al., 2022) measures the similarity of the data distribution based on two augmented samples. Unlike these methods, our proposed PBOLM achieves precise aggregation and separation of dynamic relationships among samples by employing a self-generation approach for pseudo-labels.

**Theoretical Analysis.** Arora et al. (2019) provide the first generalization bound for contrastive learning based on the Rademacher complexity. Wang & Isola (2020) analyze the contrastive learning from the perspective of alignment and uniformity on the hypersphere. Zimmermann et al. (2021b) reveal a fundamental connection between contrastive learning, generative modeling, and nonlinear independent component analysis. Wang & Isola (2020) show that the temperature parameter controls the alignment and uniformity of the learned features in the feature space. Von Kügelgen et al. (2021); Qiang et al. (2022) understand the contrastive learning from causal analysis, such as intervention and counterfactual. Ash et al. (2021) report that the performance of a representation learned via contrastive learning can degrade with the number of negative samples. However, Awasthi et al. (2022); Nozawa & Sato (2021) argue that a larger number of negative samples do not necessarily harm contrastive learning. This paper also provides a theoretical analysis of the effectiveness of the proposed method from the perspective of data generation. Liu et al. (2020) has studied the early-time training phenomenon in bounded label noise scenarios and proposes a method to exploit the use of bounded noisy labels. In this paper, we extend the bounded setting to unbounded setting, and we are the first to address this problem in SSL scenario.

## 3 PRELIMINARIES

Self-Supervised Learning (SSL) (Chen et al., 2020a; Wang & Isola, 2020) aims to learn a feature extractor $f$ that maps the original input samples from their raw space into a feature space. These features are further projected from the feature space into an embedding space through the use of a projection head, represented as $f_{ph}$. The training data is organized as a mini-batch denoted as $X_{tr} = \{x_i\}_{i=1}^N$, where $x_i$ represents the $i$-th sample, and $N$ is the total number of samples. Stochastic data augmentation techniques, such as random cropping, are applied to transform each sample $x_i$ into two augmented views, labeled as $x_i^1$ and $x_i^2$. Consequently, this results in the creation of an augmented dataset, denoted as $X_{tr}^{aug} = \{x_i^1, x_i^2\}_{i=1}^N$. Alternatively, this can be expressed as $X_{tr}^{aug} = \{^iX_{tr}^{aug}\}_{i=1}^2$, where $^iX_{tr}^{aug} = \{x_j^i\}_{j=1}^N$. Note that the samples in $X_{tr}$ serve as the ancestors of those in $X_{tr}^{aug}$. Then, we calculate the feature representations as follows: $z_i^j = f_{ph}(f(x_i^j))$, where $i \in \{1, ..., N\}$ and $j \in \{1, 2\}$. Additionally, SSL methods usually normalize these feature vectors as $z_i^j = z_i^j / ||z_i^j||_2$, ensuring that they have a unit Euclidean norm.

The prevailing SSL methods are typically structured around two fundamental components: alignment and constraint. The alignment component's objective is to maximize the similarity between the features extracted from different augmented samples that share a common ancestor. Conversely, the constraint component introduces additional prior knowledge, governing various aspects of the data distribution, parameter updates, and feature representations during the training process. Consequently, SSL methods can be unified under a common framework, as expressed below:

$$\min_{f, f_{ph}} \mathcal{L}_{\text{align}}(X_{tr}^{aug}, f, f_{ph}) + \mathcal{L}_{\text{prior}}(X_{tr}^{aug}, f, f_{ph}) \tag{1}$$

where $\mathcal{L}_{\text{align}}$ and $\mathcal{L}_{\text{prior}}$ denote the alignment and constraint losses, respectively. In the subsequent paragraph, we revisit three prominent SSL methods within the framework defined by Equation (1).

SimCLR (Chen et al., 2020a) operates as an instance-based learning method, wherein each sample during training is assigned a unique class. Notably, SimCLR, as elucidated in Theorem 1 of (Wang & Isola, 2020), enforces a constraint on the data distribution, ensuring it remains uniform. BYOL (Grill et al., 2020) is a SSL method that adopts a distinctive approach by disregarding the use of negative samples. Instead, BYOL introduces constraints on the gradient backpropagation process, which are tailored to the characteristics of the training data. Barlow Twins (Zbontar et al., 2021) presents a unique SSL method that deviates from the conventional reliance on negative samples, gradient stopping mechanisms, or the utilization of asymmetric networks. Instead, Barlow Twins enforces constraints on the feature representation by promoting the decorrelation of its vector components.

## 4 MOTIVATION AND IN-DEPTH ANALYSIS

### 4.1 PROBLEM FORMULATION

We start by motivating our method before explaining its details in Section 5. Equation (1) highlights a fundamental challenge: SSL methods encounter difficulties during training due to the absence of labeled information. This limitation hampers their ability to accurately model the dynamic relationships among augmented samples. For instance, SSL primarily focuses on the aggregation of augmented samples from the same ancestral source but overlooks the aggregation of augmented samples derived from ancestor samples with identical labels. Concurrently, SSL confines itself to imposing prior knowledge on the overall data distribution while neglecting the crucial notion that augmented samples resulting from ancestor samples with different labels should ideally exhibit a tendency to move apart from one another. One feasible solution to address the aforementioned challenges is as follows: initially, generating a pseudo-label for each augmented sample and then utilizing these pseudo-labels to guide feature representation learning.

### 4.2 LABEL SPACE INCONSISTENCY

A significant challenge in generating a pseudo-label lies in ensuring that the pseudo-label space aligns with the true label space. This challenge arises due to the unknown number of classes in the training data, making the direct application of clustering algorithms to create pseudo-labels problematic. Notably, SSL constitutes an instance-level representation learning paradigm. During SSL

training, an augmented sample is regarded as an anchor point, while another augmented sample sharing the same ancestor is treated as a positive sample relative to the anchor point. The remaining augmented samples are designated as negative samples in relation to the anchor point. SSL's objective is to cluster positive samples with anchor points while constraining the entire data distribution to satisfy a given prior. This SSL learning paradigm offers valuable insights for designing a pseudo-label generation mechanism. It suggests the creation of an instance-level pseudo-label generation mechanism that not only considers another augmented sample with the same ancestor as a positive sample but also incorporates augmented samples from different ancestors exhibiting notably high similarity above a specified threshold as positive samples. All remaining samples can be categorized as negative samples. An advantage of this approach is that, even though the generated pseudo-labels may not be entirely accurate, they guarantee complete alignment between the pseudo-label space and the true label space.

We present a specific method for generating pseudo-labels. Drawing inspiration from BYOL, it becomes evident that iteratively leveraging the output of an online network related to one augmented sample to predict the output of the target network related to another augmented sample is a potent strategy. In this context, the output of the target network effectively serves as labels for the online network's outputs. Consequently, we propose the use of feature representations obtained from the target network's output to generate pseudo-labels: for each anchor sample, samples exhibiting a similarity greater than $\tau$ are assigned a positive class label, while those falling below this threshold receive a negative class label. Furthermore, the online network can update its parameters through standard gradient backpropagation, while the parameters of the target network can be updated using the same moving average technique as employed in BYOL.

### 4.3 UNBOUNDED PSEUDO-LABEL NOISE

Then, we illustrate that the label noise in the generated noise label is unbounded, that is, the mislabeling rate for an augmented image can approach 1, e.g., $\exists A \subset X_{tr}^{aug}, \forall x_i^j \in A, \Pr[Y = b|Y = a, x_i^j] \to 1$, where $a, b \in \{+1, -1\}$. We begin with a two-component Multivariate Gaussian mixture distribution with an equal prior scenario. Without loss of generality, we denote positive-class samples and negative-class samples as Gaussian distributions with mean $\mu_1$ and $\mu_2$, both having variance $\sigma^2 I_d$, where $I_d \in \mathbb{R}^{d \times d}$. To simulate label learning based on pseudo-labels, we set up the following scenario: initially, we have a prior mixture distribution, e.g., $\mathcal{N}(\mu_1, \sigma^2 I_d)$ for positive-class samples and $\mathcal{N}(\mu_2, \sigma^2 I_d)$ for negative-class samples. We train a classifier based on this mixture distribution to assign pseudo-labels to each target sample accordingly. Subsequently, considering the similarity between the target network and the online network, we assume that the distribution of target data experiences a mean shift compared to the prior distribution, while the variance remains unchanged. For instance, $\mathcal{N}(\mu_1 + \Delta, \sigma^2 I_d)$ for positive-class samples and $\mathcal{N}(\mu_2 + \Delta, \sigma^2 I_d)$ for negative-class samples, where $\Delta$ represents the shift between the two distributions. It is worth noting that the magnitude of the shift is closely related to the complexity of the mixture distribution (Stojanov et al., 2021; Zhao et al., 2019). Specifically, $\Delta$ is positively correlated with the vector $\mu_2 - \mu_1$. Thus, we establish the following relationship characterizing label noise's characteristics:

**Theorem 4.1.** *Assume that $\Delta$ is positively correlated with the vector $\mu_2 - \mu_1$, e.g., $\Delta^{\mathrm{T}}(\mu_2 - \mu_1) > 0$. For a sample $x$ in the target data distribution with $y$ as the ground-truth label, we can obtain:*

$$\Pr[f_{cl}(x) \neq y] = \frac{1}{2}\Phi(-\frac{\delta_1}{\sigma}) + \frac{1}{2}\Phi(-\frac{\delta_2}{\sigma}) \tag{2}$$

*where $f_{cl}$ be the optimal classifier trained by prior distribution, $\delta_1 = ||\frac{\mu_2 - \mu_1}{2} - c||\mathrm{sign}(\frac{\mu_2 - \mu_1}{2} - c)$, $\delta_2 = \frac{\mu_2 - \mu_1}{2} + c$, $c = a(\mu_2 - \mu_1)$, $a = \frac{\Delta^{\mathrm{T}}(\mu_2 - \mu_1)}{||\mu_2 - \mu_1||^2}$ is the magnitude of distribution shift, and $\Phi$ is the standard normal cumulative distribution function. Meanwhile, if $x \in U$, we have:*

$$\Pr[f_{cl}(x) \neq y] \geq 1 - \delta \tag{3}$$

*where $\delta \in (0, 1)$ and $U = \{x : ||x - \mu_1 - \Delta|| \leq \frac{(d\sigma - 2\sigma \log \frac{1-\delta}{\delta})}{2\sqrt{d}}\} \cap \{x : x^{\mathrm{T}} 1_d > \frac{(d\sigma + 2\mu_1^{\mathrm{T}} 1_d)}{2}\}$. Also, if $a > \log \frac{1-\delta}{\delta}/d$, $U$ is non-empty.*

The proof is provided in the Appendix. Theorem 4.1 demonstrates that: (1) the mislabeling rate increases as the magnitude of the distribution shift increases; (2) the label noise generated by $f_{cl}$

becomes unbounded for any $x \in U$. In practical scenarios, the region U is never empty, especially when $f_{cl}$ and feature extractor $f$ are trained on high-dimensional data, where $d \gg 1$. Consequently, it becomes relatively straightforward to satisfy $\alpha \to 0$. Moreover, the probability measure on U increases as the magnitude of $a$ grows. This signifies that a greater number of data points start to gain unbounded noisy labels. When meeting the unbounded label noise, we can derive the following Lemma to illustrate that many existing methods designed to handle label noise (Ghosh et al., 2017; Wang et al., 2019; Ma et al., 2020; Englesson & Azizpour, 2021) have failed to exhibit robustness.

**Lemma 4.1.** *Assume that the loss of a classifier $f_{cl}$ under the clean label $y$ scenario be denoted as $R(f_{cl}, y) = \mathbb{E}_{x,y} \mathcal{L}(f_{cl}(x), y)$, and the loss under unbounded noisy label $y'$ scenario be denoted as $R(f_{cl}, y') = \mathbb{E}_{x,y'} \mathcal{L}(f_{cl}(x), y')$. Then the global minimizer $f_{cl}^*$ of $R(f_{cl}, y)$ disagrees with the global minimizer $f_{cl}^*$ of $R(f_{cl}, y')$ with a high probability at least $1 - \delta$.*

The proof is provided in the Appendix. Note that the $\mathcal{L}(\cdot)$ in the above Lemma refer to the noise-robust loss in Wang et al. (2019); Ghosh et al. (2017); Englesson & Azizpour (2021); Ma et al. (2020). At the same time, we need to point out that when the label noise follows the bounded assumption, e.g., $\Pr[Y = b | Y = a, x_i^j] \leq m$, where $a, b \in \{+1, -1\}$ and $0 \leq m < 1$, the methods discussed in the mentioned literature are noise tolerant, e.g., the minimizer $\tilde{h}^\star$ converges to the minimizer $f_{cl}^*$ of $R(f_{cl}, y)$ converges to the minimizer $f_{cl}^*$ of $R(f_{cl}, y')$ with a high probability.

## 4.4 INTRIGUING PROPERTY FOR LEARNING WITH UNBOUNDED NOISE LABEL

Another key challenge in the pseudo-label generation approach lies in accurately learning the dynamic relationships of aggregation and separation among augmented samples, particularly when dealing with unbounded, noisy pseudo-labels. To address this challenge systematically, we first present an intriguing property for learning with the unbounded noise label.

Based on the proposed instance-level pseudo-label generation mechanism, which is presented in Subsection 4.2, we consider a process in which all training samples are initially fed into the target network to obtain their feature representations. Then, based on anchor points, we can model the distribution of positive-class samples as a Gaussian distribution with a mean equal to the anchor sample and a covariance matrix denoted by $\sigma^2 I_d$, thus, we have $\mathcal{N}(\mu, \sigma^2 I_d)$, where $||\mu|| = 1$ denotes the normalized anchor representation. Finally, we can assign pseudo-labels to the sample $x$ based on $\beta(x) = \text{sign}(\mathbb{1}\{x^T \mu > r\} - 0.5)$, where $r > 0$ is the hyper-parameter. For example, if $\beta(x) > 0$, then $x$ is labeled as positive, otherwise, $x$ is labeled as negative.

As we delve into the dynamic process of learning with unbounded pseudo-labels, we unearth an intriguing phenomenon. Specifically, the training dynamics of the classifier exhibit a preference for fitting clean samples. Consequently, the classifier demonstrates higher prediction accuracy for mislabeled samples during the early training stages. These training characteristics can be highly advantageous, especially in pseudo-label learning scenarios with unbounded label noise.

Given the unbounded label noise data $\{x_i, \tilde{y}_i\}_{i=1}^{2n}$, where $\tilde{y}_i$ denotes the noisy label, we can model the training dynamics of gradient descent on the online network $f$ with the following objective:

$$f_{t+1} = f_t - \eta \nabla_f \mathcal{L}(f_t)$$
$$s.t. \mathcal{L}(f_t) = \frac{1}{n} \sum_{i=1}^{2n} \sum_{j=1, j \neq i}^{2n} \log(1 + \exp(-\tilde{y}_j f_t(\mu_i)^T f_t(x_j))) \tag{4}$$

where $\eta$ is the learning rate and $\mu_i$ is the representation of the anchor sample. Then, we use the following theorem to build the connection between the prediction accuracy for mislabeled samples at an early-training time $T$.

**Theorem 4.2.** *Let $B = \{x : \tilde{y} \neq y\}$ represent a set of mislabeled samples. We define $\kappa(B; f)$ as the prediction accuracy calculated using ground-truth labels and the labels predicted by a classifier with parameter $f$ for mislabeled samples. If, at most, half of the samples are mislabeled ($r < 1$), then there exists a specific time point $T$ and a constant $c_0 > 0$ such that for any $0 < \sigma < c_0$ and as $n$ approaches infinity, with probability $1 - o_p(1)$: $\kappa(B; f_T) \geq 1 - \exp\{-\varsigma g(\sigma)^2\}$, where $\varsigma > 0$ is a constant, $g(\sigma) = (\sqrt{2\pi} F[\frac{1-r}{\sqrt{2}r}] + 2\sigma \exp(-\frac{(r-1)^2}{2\sigma^2}))/(2\sqrt{2\pi}(1 + 2\sigma)\sigma) > 0$ is a monotone decreasing function that $g(\sigma) \to \infty$ as $\sigma \to 0$, and $F[x] = \frac{2}{\sqrt{\pi}} \int_0^x e^{-t^2} dt$.*

The proof is provided in the Appendix. Theorem 4.2 establishes that at a specific time point $T$, the $f$ trained using the gradient descent algorithm can offer accurate predictions for mislabeled samples. This accuracy is lower bounded by a function related to the variance of clusters $\sigma$. As $\sigma \to 0$, the predictions for all mislabeled samples converge to their ground-truth labels, denoted as $\kappa(B; f_T) \to 1$. However, as the classifier undergoes extended training, it gradually memorizes the mislabeled data. Consequently, the predictions for mislabeled samples deviate from their ground-truth labels and align with their incorrect labels instead. Drawing from these insights, it becomes evident that the memorization of mislabeled data can be mitigated by utilizing their predicted labels during the early stages of training.

Building upon the characteristics we have uncovered during the learning process, we propose the following learning objectives, called Precise Adjustment Regularization (PAR), to address unbounded label noise in pseudo-labels:

$$\mathcal{L}_{\text{PAR}}(f) = \frac{1}{n} \sum_{i=1}^{2n} \sum_{j=1, j \neq i}^{2n} \log(1 - \tilde{y}_j \text{La}(f)) \tag{5}$$

where $\text{La}(f) = (\text{out}(f), 1 - \text{out}(f))$, $\text{out}(f) = f(\mu_i)^{\text{T}} f(x_j) / \sum_{j=1, j \neq i}^{2n} f(\mu_i)^{\text{T}} f(x_j)$, and $\tilde{y}_j \in \{(1, 0), (0, 1)\}$. Equation (5) encourages model predictions to stick to the early-time predictions for $x$. Also, we overload $f(\mu_i)^{\text{T}} f(x_j) / \sum_{j=1, j \neq i}^{2n} f(\mu_i)^{\text{T}} f(x_j)$ to be the probabilistic output for the sample $x_j$ related to the anchor $\mu_i$, and $\tilde{y}_j(t) = \beta \tilde{y}_j(t-1) + (1 - \beta) \text{out}(f_t)$ is the moving average prediction for x, where $\beta$ is a hyperparameter. To see how PAR prevents the model from memorizing the label noise, we calculate the gradient of Equation (5) with respect to $\text{out}(f_t)$:

$$\frac{\partial \mathcal{L}_{\text{PAR}}(f_t)}{\partial \text{out}(f_t)} = -\sum_{i=1}^{2n} \sum_{j=1, j \neq i}^{2n} \frac{\tilde{y}_j(t)}{1 - \tilde{y}_j(t)^{\text{T}} \text{out}(f_t)} \tag{6}$$

It's important to note that minimizing Equation (5) encourages $\text{out}(f_t)$ to closely align with $\tilde{y}_j(t)$. When $\tilde{y}_j(t)$ aligns better with $\text{out}(f_t)$, the gradient's magnitude increases, causing the gradient for aligning $\text{out}(f_t)$ with $\tilde{y}_j(t)$ to dominate over the gradients for other loss terms that align $\text{out}(f_t)$ with noisy labels. As training progresses, the moving averaged predictions $\tilde{y}_j(t)$ for target samples gradually approach their ground-truth labels until time $T$. Consequently, Equation (5) serves the crucial purpose of preventing the model from memorizing label noise. It achieves this by compelling the model's predictions to remain close to these moving averaged predictions $\tilde{y}_j(t)$, which are highly likely to represent ground-truth labels.

## 5 METHODOLOGY

In this section, we propose a novel mechanism called PAR-based bi-level optimization learning mechanism (PBOLM) to induce better representation learning through SSL. We also analyze the proposed method from the point of view of data generation and prove its effectiveness.

### 5.1 PAR-BASED BI-LEVEL OPTIMIZATION LEARNING MECHANISM

PBOLM consists of two modules: the SSL module and the pseudo-label learning (PLL) module. The SSL module comprises an online network $f$ (feature extractor) and a projection head $f_{ph}$. The PLL module is the implementation of the instance-level pseudo-label generation mechanism and consists of a target network (another feature extractor) $f_t$ and a projection head $f_{ph}^t$. Given the training dataset $X_{tr}^{aug} = \{x_i^1, x_i^2\}_{i=1}^N$, we first input them into $f_t$ and $f_{ph}^t$ to obtain the embeddings, e.g., $z_i^j = f_{ph}(f_t^t(x_i^j))$ and $z_i^j = z_i^j / ||z_i^j||_2$. Then, given an anchor, the PLL module assigns pseudo-labels to all samples in the training set except for the anchor based on the similarity between the anchor and the sample embeddings. Specifically, if the anchor is denoted as $z_i^j$, and if $\beta(x) = \text{sign}(\mathbb{1}\{z^{\text{T}} z_i^j > r\} - 0 > 0$, it assigns the labels $(1, 0)$ to $z$; otherwise, it assigns the labels $(0, 1)$ to $z$, where $z \in \{z_i^1, z_i^2\}_{i=1}^N$ and $z \neq z_i^j$. The updates of $f_t$ and $f_{ph}^t$ are consistent with the target network update method in BYOL. Once we obtain a series of pseudo-labels for each anchor, we proceed with SSL methods.

Drawing inspiration from BYOL, a well-designed target network can induce better learning in the online network. Motivated by this, we propose utilizing PBOLM to induce better learning in SSL. Given the pseudo-labels generated by the PLL module, PBOLM learns $f$ and $f_{ph}$ as follows:

$$
\begin{aligned}
&\min_{f, f_{ph}} \mathcal{L}_{\text{align}}(X_{tr}^{aug}, f, f_{ph}) + \mathcal{L}_{\text{prior}}(X_{tr}^{aug}, f, f_{ph}) \\
&s.t. \min_{f, f_{ph}} \mathcal{L}_{\text{PAR}}(f_{ph} \cdot f)
\end{aligned}
\tag{7}
$$

where $f_{ph} \cdot f = f_{ph}(f(\cdot))$. The optimization process of Equation (7) can be divided into two steps. In the first step, we obtain $f_t$ and $f_{ph}^t$ by:

$$
f_t = f_{t-1} - \eta \nabla_f \mathcal{L}_{\text{PAR}}(f_{ph}^{t-1} \cdot f_{t-1}), f_{ph}^t = f_{ph}^{t-1} - \eta \nabla_f \mathcal{L}_{\text{PAR}}(f_{ph}^{t-1} \cdot f_{t-1})
\tag{8}
$$

where $\eta$ is the learning rate, $f_0 = f$, and $f_{ph}^0 = f_{ph}$. In the second step, we update $f$ and $f_{ph}$ by:

$$
\begin{aligned}
&f = f - \eta \nabla_f \mathcal{L}_{\text{ssl}}(f_{ph}^t, f_t), f_{ph} = f_{ph} - \eta \nabla_f \mathcal{L}_{\text{ssl}}(f_{ph}^t \cdot f_t) \\
&s.t. \quad \mathcal{L}_{\text{ssl}}(f_{ph}^t, f_t) = \mathcal{L}_{\text{align}}(X_{tr}^{aug}, f_{ph}^t, f_t) + \mathcal{L}_{\text{prior}}(X_{tr}^{aug}, f_{ph}^t, f_t)
\end{aligned}
\tag{9}
$$

Upon closer examination of Equation (7) from a more granular perspective, it can be observed that $f_t, f_{ph}^t$ is obtained by minimizing $\mathcal{L}_{\text{PAR}}(f_{ph} \cdot f)$. Therefore, $f_t$ and $f_{ph}^t$ have already distilled the dynamic information from unbounded noisy labels. The second optimization step can be understood as follows: The alterations of $f$ and $f_{ph}$ lead to the corresponding changes in $f_t$ and $f_{ph}^t$, consequently impacting the overall value of the $\mathcal{L}_{\text{ssl}}(f_{ph}^t, f_t)$. It's essential to note that $\mathcal{L}_{\text{ssl}}(f_{ph}^t, f_t)$ can only reach its minimum when $f_t$ and $f_{ph}^t$ attain an appropriate value. Hence, minimizing $\mathcal{L}_{\text{ssl}}(f_{ph}^t, f_t)$ is built upon the foundation of $f_t$ and $f_{ph}^t$. In the end, we can understand Equation (7) from two levels. The first level (constraint condition) aims to use unbounded noisy labels to accurately learn the dynamic relationships of aggregation and separation among augmented samples. The second level (objective function) further restricts the first level, aiming to constrain its behavior, that is, it should be able to facilitate a further reduction in the loss of SSL methods, thereby promoting better learning in SSL methods.

## 5.2 THEORETICAL ANALYSIS FROM A DATA GENERATION PERSPECTIVE

This subsection is presented based on SimCLR. According to nonlinear ICA (Hyvrinen & Pajunen, 1999; Zimmermann et al., 2021a; Hyvarinen & Morioka, 2017), we assume that the observations $x \to \mathcal{X}$ are generated by an invertible (i.e., injective) generative process $g : \mathcal{Z} \to \mathcal{X}$, where $\mathcal{X} \subseteq \mathbb{R}^K$ is the space of observations and $\mathcal{Z} \subseteq \mathbb{R}^N$ with $N \leq K$ denotes the space of latent factors. Influenced by the commonly used feature normalization in SimCLR, we further assume that $\mathcal{Z}$ is the unit hypersphere $\mathbb{S}^{N-1}$. Additionally, we assume that the ground-truth marginal distribution of the latents of the generative process is uniform and that the conditional distribution (under which positive pairs have high density) is a von Mises-Fisher (vMF) distribution:

$$
p(z) = |\mathcal{Z}|^{-1}, p(z|\tilde{z}) = C_p^{-1} e^{kz^{\mathrm{T}}\tilde{z}}, C_p = \int e^{kz^{\mathrm{T}}\tilde{z}} d\tilde{z} = \text{const}, x = g(z), \tilde{x} = g(\tilde{z})
\tag{10}
$$

where $k$ is a hyper-parameter. From the perspective of nonlinear ICA, we are interested in understanding how the representations $f(x)$ which minimize the contrastive loss are related to the ground-truth source signals $z$. To study this relationship, we focus on the map $h = f \cdot g$ between the recovered source signals $h(z)$ and the true source signals $z$. Then, we have:

**Theorem 5.1.** *Assume $\mathcal{Z}$ is the unit hypersphere $\mathbb{S}^{N-1}$ and Equation (10) is true. Let $g$ be injective and $h$ be differentiable. Given an anchor $z$ in $X_{tr}^{aug}$, if we can model $P(\cdot|z)$ correctly, $f$ is differentiable and minimizes the contrastive loss as defined in SimCLR, we can obtain that when $N \to \infty$, $h = f \cdot g$ is linear, i.e., $f$ recovers the latent sources up to an orthogonal linear transformation and a constant scaling factor.*

PBOLM generates pseudo-labels for each augmented sample and utilizes these pseudo-labels to guide SSL in better capturing the aggregation and separation relationships among the data points. Therefore, compared to SimCLR, PBOLM is better at accurately modeling $P(\cdot|z)$ given the anchor $z$. In other words, PBOLM can recover the latent sources more than SimCLR.

Table 1: Classification accuracy for small, medium, and large datasets. The backbone is ResNet-18 for the first four datasets and ResNet-50 for the last two datasets.

| Methods | CIFAR-10 | | CIFAR-100 | | STL-10 | | Tiny ImageNet | | ImageNet-100 | | ImageNet | |
|---|---|---|---|---|---|---|---|---|---|---|---|---|
| | linear | 5-nn | linear | 5-nn | linear | 5-nn | linear | 5-nn | top-1 | top-5 | top-1 | top-5 |
| SimCLR Chen et al. (2020a) | 91.80 | 88.42 | 66.83 | 56.57 | 90.51 | 85.68 | 48.82 | 32.86 | 70.15 | 89.75 | 69.32 | 89.15 |
| BYOL Grill et al. (2020) | 91.73 | 89.26 | 66.60 | 56.82 | 91.86 | 88.61 | 51.01 | 36.14 | 74.89 | 92.83 | 74.31 | 91.62 |
| BarlowTwins Zbontar et al. (2021) | 90.88 | 88.78 | 66.67 | 56.39 | 90.71 | 86.31 | 49.74 | 33.61 | 72.88 | 90.99 | 73.22 | 91.01 |
| SimSiam Chen & He (2021) | 91.51 | 89.31 | 66.73 | 56.87 | 91.92 | 88.54 | 50.92 | 35.98 | 74.78 | 92.84 | 71.33 | - |
| W-MSE Ermolov et al. (2021) | 91.99 | 89.87 | 67.64 | 56.45 | 91.65 | 88.49 | 49.22 | 35.44 | 75.33 | 92.78 | 72.56 | - |
| SwAV Caron et al. (2020) | 90.17 | 86.45 | 65.23 | 54.77 | 89.12 | 84.12 | 47.13 | 31.07 | 75.77 | 92.86 | 75.30 | - |
| SSL-HSIC Li et al. (2021) | 91.95 | 89.91 | 67.22 | 57.01 | 92.06 | 88.87 | 51.42 | 36.03 | 74.77 | 92.56 | 72.13 | 90.33 |
| VICReg Bardes et al. (2022) | 91.08 | 88.93 | 66.91 | 56.47 | 91.11 | 86.24 | 50.17 | 34.24 | 74.88 | 92.84 | - | - |
| SimCLR* | 92.03 | 89.14 | 66.95 | 56.95 | 90.71 | 85.88 | 49.18 | 33.12 | 70.47 | 89.89 | 70.01 | 89.24 |
| BYOL* | 91.89 | 90.02 | 66.91 | 56.92 | 91.93 | 88.81 | 51.17 | 36.22 | 74.92 | 92.91 | 74.53 | 91.22 |
| Barlow Twins* | 90.91 | 88.88 | 66.94 | 56.57 | 90.97 | 86.61 | 50.06 | 34.01 | 72.92 | 91.21 | 73.68 | 91.25 |
| SimCLR + PBOLM | 92.95 | 90.34 | **67.99** | 57.35 | 92.89 | 87.99 | 51.21 | 36.42 | 73.89 | 91.97 | 73.99 | 91.96 |
| BYOL + PBOLM | **92.99** | **90.95** | 67.91 | **57.41** | **93.99** | **89.99** | **52.61** | **37.71** | **75.99** | **93.99** | **75.99** | **92.99** |
| Barlow Twins + PBOLM | 92.64 | 90.81 | 67.46 | 56.99 | 93.35 | 88.71 | 51.94 | 36.55 | 74.95 | 92.34 | 74.94 | 91.99 |

Table 2: Semi-supervised classification. We finetune the pre-trained model using 1% and 10% training samples of ImageNet.

| Methods | Epochs | 1% | | 10% | |
|---|---|---|---|---|---|
| | | top-1 | top-5 | top-1 | top-5 |
| SimCLR | 1000 | 48.3 | 75.5 | 65.6 | 87.8 |
| BYOL | 1000 | 53.2 | 78.4 | 68.8 | 89.1 |
| SwAV | 1000 | 53.9 | 78.5 | 70.2 | 89.9 |
| BarlowTwins | 1000 | 54.9 | 79.4 | 69.6 | 89.1 |
| SimCLR + PBOLM | 1000 | 53.7 | 78.3 | 68.4 | 89.1 |
| BYOL + PBOLM | 1000 | 55.9 | 78.8 | 70.1 | 89.9 |
| Barlow Twins + PBOLM | 1000 | **56.9** | **78.9** | **71.1** | **90.1** |

Table 3: The results of transfer learning on object detection and instance segmentation with C4-backbone as the feature extractor.

| Methods | Object Det. | | | Instance Seg. | | |
|---|---|---|---|---|---|---|
| | AP | $AP_{50}$ | $AP_{75}$ | AP | $AP_{50}$ | $AP_{75}$ |
| SimCLR | 37.9 | 57.7 | 40.9 | 33.2 | 54.6 | 35.3 |
| SwAV | 37.6 | 57.6 | 40.2 | 33.0 | 54.2 | 35.1 |
| BYOL | 37.9 | 57.8 | 40.9 | 33.1 | 54.3 | 35.0 |
| SimSiam | 37.9 | 57.5 | 40.9 | 33.3 | 54.2 | 35.2 |
| BarlowTwins | 39.2 | 59.0 | 42.5 | 34.2 | 56.0 | 36.5 |
| SimCLR + PBOLM | 39.3 | 59.9 | 42.7 | 35.1 | 56.2 | 37.7 |
| BYOL + PBOLM | **40.7** | 59.6 | 43.2 | **35.9** | 56.7 | 36.5 |
| Barlow Twins + PBOLM | 39.6 | **60.1** | **43.7** | 35.5 | **57.0** | **38.1** |

# 6 EXPERIMENTS

## 6.1 EVALUATION ON BENCHMARK DATASET

**Benchmark Dataset**. For classification task, we evaluate PBOLM on six image datasets, including CIFAR-10 and CIFAR-100 datasets (Krizhevsky et al., 2009), STL-10 dataset (Coates et al., 2011), Tiny ImageNet dataset (Le & Yang, 2015), ImageNet-100 dataset (Tian et al., 2020), and ImageNet dataset (Krizhevsky et al., 2012). For transfer learning task, we validate PBOLM on the instance segmentation and object detection tasks on the COCO (Lin et al., 2014) dataset.

**Unsupervised Learning**. Experiments follow the most common evaluation protocol for self-supervised learning. A Stochastic Gradient Descent (SGD) with a momentum of 0.9 to minimize our objective functions, the linear classifier is trained for 500 epochs with a mini-batch of 256. The initial learning rate is set to $10^{-2}$, which decays to $5 \times 10^{-6}$ until the training is completed. **Results.** Table 1 shows the results of the linear classifier and 5-nn classifier on four small-scale datasets, e.g., CIFAR-10, CIFAR-100, STL-10, and Tiny Imagenet datasets, the top-1 and top-5 classification accuracy on a medium-scale dataset, and the top-1 and top-5 classification accuracy on a larger-scale dataset, e.g., the ImageNet dataset. We can observe that the classification accuracy of the proposed PBOLM outperforms other state-of-the-art methods. We observe that the results of PBOLM outperform other methods by a relative margin. Therefore, we can obtain that PBOLM is effective.

**Semi-Supervised Learning**. Experiments follow the most common evaluation protocol for semi-supervised learning Zbontar et al. (2021). We sample 1% or 10% of the training datasets as the labeled data in a class-balanced way. We fine-tune the models on these two subsets for 50 epochs with a classifier learning rate 1.0 (0.1), and backbone learning rate 0.0001 (0.01) for the 1% (10%) subset. **Results**. Table 2 reports the classification results obtained on the ImageNet compared with existing methods using two pre-trained models with 1000 epochs. From the results, we can observe that the classification accuracy of the proposed PBOLM outperforms other state-of-the-art methods by more than 1%. These results demonstrate the effectiveness of the proposed method.

**Transfer Learning**. Experiments follow the common setting for transfer learning used by existing methods(Zbontar et al., 2021; Grill et al., 2020). We evaluate PBOLM on object detection and

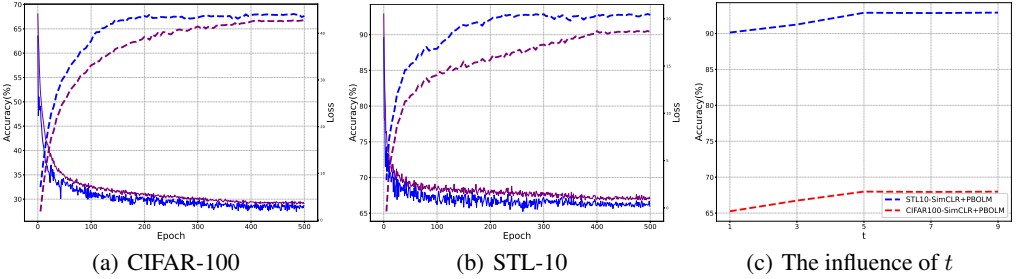

(a) CIFAR-100       (b) STL-10       (c) The influence of $t$

Figure 1: Based on CIFAR-100 and STL-10 datasets, (a) and (b) show the variation curves of testing accuracy and loss magnitude. (c) shows the impact of the hyperparameter $t$ on SimCLR + PBOLM.

instance segmentation tasks on COCO (Lin et al., 2014) datasets. We use Mask R-CNN(He et al., 2017) with a $1\times$ schedule and the same backbone as Faster R-CNN. **Results**. We report the results of our proposed method compared with baseline methods in Table 3, showing that the proposed method brings performance improvements on different downstream tasks.

## 6.2 ABLATION STUDY

PBOLM is trained with a bi-level optimization mechanism. An inherent challenge is whether it is feasible to directly train PBOLM with objective $\mathcal{L}_{SSL} + \mathcal{L}_{PAR}$. To address this question, we directly optimize the SSL method based on the objective $\mathcal{L}_{SSL} + \mathcal{L}_{PAR}$, and we use $*$ denote this kind of trained SSL method, e.g., SimCLR*. The final results are summarized in Table 1. We can observe that the test accuracy of SSL methods with $*$ surpasses that of directly training the SSL method, yet it falls short of the test accuracy achieved by the $+$ PBOLM approach. This suggests that PAR is effective both in a direct training manner and a bi-level training manner. Also, one possible reason for PBOLM performance being superior to $*$ performance could be attributed to the bi-level optimization mechanism that can promote better learning of SSL methods.

We have conducted experiments using the CIFAR-100 and STL-10 datasets, tracking the changes in contrastive loss and test accuracy during the training process for both SimCLR and Sim-CLR+PBOLM. The results are depicted in Figure 1. It is evident from the training phase that Sim-CLR+PBOLM exhibits a significantly faster convergence rate than SimCLR, as observed in both loss values and training accuracy. One possible explanation for this phenomenon is that the proposed PAR encourages the feature extractor to learn a better initialization of parameters, which subsequently accelerates model convergence and enhances its performance during contrastive learning. This also validates the effectiveness of the PAR-based bi-level optimization learning mechanism.

From Equation 8, we can deduce that the first step of PBOLM training requires updating for $t$ times. As per Theorem 4.2, an appropriate choice of $t$ can enable PAR to learn accurately even under unbounded noisy labels. However, exceeding a certain time limit, PAR starts overfitting the noisy labels, thereby reducing the model's generalization. To explore the impact of $t$, we conducted experiments using the CIFAR-100 and STL-10 datasets, considering different values of $t$ from the set $1, 3, 5, 7, 9$. The results are presented in Figure 6, showing that an appropriate choice of $t$ does influence the model's final performance. This also validates the correctness of Theorem 4.2.

## 7 CONCLUSIONS

This paper primarily addresses the limitation in SSL methods, which struggle to model the dynamic relationships where similar augmented samples should cluster together and dissimilar ones should separate. To tackle this, we propose an instance-level pseudo-label generation mechanism. Then, PAR is proposed to accurately learn the aggregation and separation relationships among different augmented samples using pseudo-labels. Subsequently, we present PBOLM, which incorporates PAR into the SSL training process. Finally, we offer a theoretical perspective on data generation to show that PBOLM can learn comprehensive data generation factors and validate its effectiveness through various downstream tasks.

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

# A  PROOFS FOR THEOREM 4.1

*Proof.* The Bayes classifier $f_S$ predicts $x$ to the first component when

$$\log \frac{\Pr[y = 1|X = x]}{\Pr[y = -1|X = x]} > 0. \tag{11}$$

Given by $\mathcal{N}(\mu_1, \sigma^2 I_d)$ and $\mathcal{N}(\mu_2, \sigma^2 I_d)$, respectively. Based on Bayes' rule, Eq. (11) equals to:

$$\log \frac{\Pr[X = x|y = 1]}{\Pr[X = x|y = -1]} > 0 \tag{12}$$

Then, we can obtain:

$$h_S(x) = \log \frac{\Pr[X = x|y = 1]}{\Pr[X = x|y = -1]} = \frac{x^\top (\mu_1 - \mu_2)}{\sigma^2} - \frac{\|\mu_1\|^2 - \|\mu_2\|^2}{2\sigma^2}. \tag{13}$$

When $\Delta = 0$, the mislabeling rate related to the Bayes error is given by:

$$\Pr_{(x,y)}[f_S(x) \neq y] = \frac{1}{2} \Pr_{x \sim \mathcal{N}(\mu_1, \sigma^2 I_d)}[h_S(x) < 0|y = 1] + \frac{1}{2} \Pr_{x \sim \mathcal{N}(\mu_2, \sigma^2 I_d)}[h_S(x) > 0|y = -1] \tag{14}$$

Then, we have:

$$\Pr_{(\mathbf{x},y)}[f_S(\mathbf{x}) \neq y] = \Phi(-\frac{\|\mu_2 - \mu_1\|}{2\sigma}). \tag{15}$$

Consider the case when $\Delta \neq \mathbf{0}$. The distributions of the first and the second component are $\mathcal{N}(\mu_1 + \Delta, \sigma^2 I_d)$ and $\mathcal{N}(\mu_2 + \Delta, \sigma^2 I_d)$, respectively. We project $\Delta$ onto the vector $\mu_2 - \mu_1$ to get the component of $\Delta$ that is perpendicular to the hyperplane, which is given by:

$$\mathbf{c} = (\mu_2 - \mu_1) \frac{\Delta^\top (\mu_2 - \mu_1)}{\|\mu_2 - \mu_1\|^2}. \tag{16}$$

Note that the results also hold for the case where $\Delta$ is negatively correlated to $\mu_2 - \mu_1$. The whole proof can be obtained by following the very similar proof steps for the positively correlated case.

The mislabeling rate of the optimal classifier $f_S$ on target data is:

$$\Pr_{(x,y)}[f_S(x) \neq y] = \frac{1}{2} \Pr_{\mathcal{N}(\mu_1 + \Delta, \sigma^2 I_d)}[h_S(x) < 0|y = 1] + \frac{1}{2} \Pr_{\mathcal{N}(\mu_2 + \Delta, \sigma^2 I_d)}[h_S(x) > 0|y = -1] \tag{17}$$

Then, we have:

$$\Pr_{(\mathbf{x},y)}[f_S(\mathbf{x}) \neq y] = \frac{1}{2}\Phi(-\frac{d_1}{\sigma}) + \frac{1}{2}\Phi(-\frac{d_2}{\sigma}). \tag{18}$$

Without loss of generality, we choose to assume $\boldsymbol{\mu_2} = \boldsymbol{\mu_1} + \sigma \mathbf{1}_d$ as the convenient way to present our results.

Let $f_T$ be the optimal Bayes classifier for the target data. The equation $h_T(\mathbf{x_0}) = 0$ implies that

$$\Pr_{(\mathbf{x},y) \sim \mathcal{D}_T}[y = 1|X = \mathbf{x}_0] = \Pr_{(\mathbf{x},y) \sim \mathcal{D}_T}[y = -1|X = \mathbf{x}_0].$$

Note that $x_0$ is on the affine hyperplane $z$ where $h_T(z) = 0$. Any data points on this hyperplane will have the equal probabilities to be correctly classified. We start from this hyperplane and calculate another point $x_1$, where $\Pr_{(x,y) \sim \mathcal{D}_T}[y = 1|X = x_1]$ is at least $\frac{1-\delta}{\delta} \Pr_{(x,y) \sim \mathcal{D}_T}[y = -1|X = x_1]$. Thus, for any points that are mislabeled and far away from $x_1$ will result in $\Pr_{(x,y) \sim \mathcal{D}_T}[y = 1|X = x_1] \geq 1 - \delta$. We first aim to find such a data point $x_1$. Let $x_1 = x_0 - m_0 \sigma \mathbf{1}_d$, where $m_0$ is the scalar measures the distance between the point $x_1$ to the hyperplane $z$. We need to find $m_0$ such that

$$\frac{P_T(x_1|y = 1)}{P_T(x_1|y = -1)} \geq 1 - \delta, \tag{19}$$

Then, we get $m_0 \geq (\log \frac{1-\delta}{\delta})/d$. Since the isotropic Gaussian random vectors has the rotationally symmetric property, we can transform the integration of multivariate normal distribution to standard normal distribution with different intervals of integration.

The region $R_1$ is valid when data dimension $d$ is large. This is realistic in practice. Since neural networks are usually dealing with high-dimension data, for example $d \gg (1)$, the region $\mathbf{R}_1$ is valid.

We take the intersection of $\mathbf{R}_1$ and $\mathbf{R}_2$, all data points from this intersection are (1) having at least $1-\delta$ probability coming from the first component, and (2) being classified to the second component. Formally, for $(\mathbf{x}, y) \sim \mathcal{D}_T$, if $\mathbf{x} \in \mathbf{R}_1 \bigcap \mathbf{R}_2$, then

$$\Pr[f_S(\mathbf{x}) \neq y] \geq 1 - \delta, \tag{20}$$

Since $\mathbf{x}_1$ is chosen from $\mathbf{R}_1$, to verify that $\mathbf{R}_1 \bigcap \mathbf{R}_2$ is non-empty, we only need to verify that $\mathbf{x}_1$ also belongs to $\mathbf{R}_2$.

$\mathbf{x}_1 \in \mathbf{R}_2$ if and only if:

$$\mathbf{x}_1^\top \mathbf{1}_d > \frac{\sigma d + 2\boldsymbol{\mu}_1^\top \mathbf{1}_d}{2}$$

$$(\boldsymbol{\mu}_1 + \mathbf{c} + \frac{\sigma}{2}\mathbf{1}_d - m_0\sigma\mathbf{1}_d)^\top \mathbf{1}_d > \frac{\sigma d + 2\boldsymbol{\mu}_1^\top \mathbf{1}_d}{2}$$

$$(\boldsymbol{\mu}_1 + \alpha\sigma\mathbf{1}_d + \frac{\sigma}{2}\mathbf{1}_d - m_0\sigma\mathbf{1}_d)^\top \mathbf{1}_d > \frac{\sigma d + 2\boldsymbol{\mu}_1^\top \mathbf{1}_d}{2}$$

$$(\alpha - m_0)\sigma d > 0,$$

Therefore, if $\alpha > m_0 \geq (\log \frac{1-\delta}{\delta})/d$, $\mathbf{R}_1 \bigcap \mathbf{R}_2$ is non-empty.

Next, we show $\Pr_{(\mathbf{x}, y) \sim \mathcal{D}_T}[\mathbf{x} \in \mathbf{R}]$ increases as $\alpha$ increases.

Let event $\mathbf{A}_0$ be a set of $\mathbf{x}$ such that they are mislabeled by $f_S$ (i.e. $f_S(\mathbf{x}) \neq y$). Let event $\mathbf{A}_1$ be a set of $\mathbf{x}$ such that they are from the first component but are mislabeled to the second component with a probability $\Pr[f_S(\mathbf{x} \neq y)] < 1 - \delta$. Let event $\mathbf{A}_2$ be a set of $\mathbf{x}$ such that they are from the second component but are mislabeled to the first component with a probability $\Pr[f_S(\mathbf{x} \neq y)] < 1 - \delta$. Thus

$$\Pr_{(\mathbf{x}, y) \sim \mathcal{D}_T}[\mathbf{x} \in \mathbf{R}] = \Pr_{(\mathbf{x}, y) \sim \mathcal{D}_T}[\mathbf{A}_0] - \Pr_{(\mathbf{x}, y) \sim \mathcal{D}_T}[\mathbf{A}_1] - \Pr_{(\mathbf{x}, y) \sim \mathcal{D}_T}[\mathbf{A}_2] \tag{21}$$

Let event $\mathbf{A}_3$ be a set of $\mathbf{x}$ such that they are from the first component such that $\Pr[f_S(\mathbf{x} \neq y)] < 1 - \delta$ or $\Pr[f_S(\mathbf{x} = y)] < 1 - \delta$. Let event $\mathbf{A}_4$ be a set of $\mathbf{x}$ such that they are from the second component but are mislabeled to the first component. So when $\alpha$ increases, $\Pr_{(\mathbf{x}, y) \sim \mathcal{N}(\boldsymbol{\mu_2} + \boldsymbol{\Delta}, \sigma^2 \mathbf{I}_d)}[\mathbf{A}_4]$ decreases.

Since $\mathbf{A}_1 \subseteq \mathbf{A}_3$ and $\mathbf{A}_2 \mathbf{A}_4$, the probability measure on $\mathbf{R}$ is given by:

$$\Pr_{(\mathbf{x}, y) \sim \mathcal{D}_T}[\mathbf{x} \in \mathbf{R}] = \Pr_{(\mathbf{x}, y) \sim \mathcal{D}_T}[\mathbf{A}_0] - \Pr_{(\mathbf{x}, y) \sim \mathcal{D}_T}[\mathbf{A}_1] - \Pr_{(\mathbf{x}, y) \sim \mathcal{D}_T}[\mathbf{A}_2]$$

$$\geq \Pr_{(\mathbf{x}, y) \sim \mathcal{D}_T}[\mathbf{A}_0] - \Pr_{(\mathbf{x}, y) \sim \mathcal{D}_T}[\mathbf{A}_3] - \Pr_{(\mathbf{x}, y) \sim \mathcal{D}_T}[\mathbf{A}_4], \tag{22}$$

where the first term is the mislabeling rate that increases as $\alpha$ increases; the second term is a constant; the third term decreases as as $\alpha$ increases. The equality in Eq. (22) holds when $\alpha \to \infty$. Therefore, when the magnitude of the domain shift $\alpha$ increases, the lower bound of $\Pr_{(\mathbf{x}, y) \sim \mathcal{D}_T}[\mathbf{x} \in \mathbf{R}]$ increases, which forces more points to break the conventional LLN assumption. $\square$

## B  Proofs for Lemma 4.1

*Proof.* Let $\eta_{yk}(\mathbf{x})$ be the $\Pr[\tilde{Y} = k | Y = y, X = \mathbf{x}]$ probability of observing a noisy label $k$ given the ground-truth label $y$ and a sample $\mathbf{x}$. Let $\eta_y(\mathbf{x}) = \sum_{k \neq y} \eta_{yk}(\mathbf{x})$.

To let $\ell_{\text{LLN}}(\tilde{h}^\star(\mathbf{x}), k) \geq \ell_{\text{LLN}}(h^\star(\mathbf{x}), k)$ holds for all inputs $\mathbf{x}$, previous studies assume the bounded label noise, which is given by

$$1 - \eta_y(\mathbf{x}) - \eta_{yk}(\mathbf{x}) > 0 \ \forall \mathbf{x} \ \text{s.t.} \ P(X = \mathbf{x}) > 0. \tag{23}$$

For random label noise which assumes that the mislabeling probability from the ground-truth label to any other label is the same for all inputs, i.e. $\eta_{ji}(\mathbf{x}) = a_0 \, \forall i \neq j$, where $a_0$ is a constant. Let $\eta = (K-1)a_0$, then Eq. (23) is degraded to

$$1 - \eta - \frac{\eta}{K-1} > 0$$
$$1 > \frac{K}{K-1}\eta$$
$$\eta < 1 - \frac{1}{K}.$$

This bounded assumption is commonly assumed by Wang et al. (2019) (Theorem 1 in Ghosh et al. (2017), Theorem 1 in Wang et al. (2019), Lemma 1 in Ma et al. (2020) and Theorem 1 in Englesson & Azizpour (2021)).

For class-conditional label noise, which assumes the $\eta_{ji}(\mathbf{x}_1) = \eta_{ji}(\mathbf{x}_2)$ for any inputs $\mathbf{x}_1$ and $\mathbf{x}_2$. Let $\eta_{ji}(\mathbf{x}) = \eta_{ji}$, Then the bounded assumption Eq. (23) is degraded to

$$\eta_{yk} < 1 - \eta_y.$$

This bounded assumption is also commonly assumed, and it can be found in Theorem 2 in Ghosh et al. (2017), Theorem 1 in Wang et al. (2019), 2 in Ma et al. (2020) and Theorem 2 in Englesson & Azizpour (2021).

However, in SFDA, we proved that the following event $\mathbf{B}$ holds with a probability at least $1 - \delta$:

$$1 - \eta_y(\mathbf{x}) - \eta_{yk}(\mathbf{x}) < 0 \, \forall \mathbf{x} \in \mathbf{R}. \tag{24}$$

Given the result in Eq. (24), we have

$$\ell_{\mathrm{LLN}}(\tilde{h}^\star(\mathbf{x}), k) \leq \ell_{\mathrm{LLN}}(h^\star(\mathbf{x}), k).$$

When the event $\mathbf{B}$ holds, the condition $\ell_{\mathrm{LLN}}(\tilde{h}^\star(\mathbf{x}), k) \leq \ell_{\mathrm{LLN}}(h^\star(\mathbf{x}), k)$ holds.

Note that only $\ell_{\mathrm{LLN}}(\tilde{h}^\star(\mathbf{x}), k) \geq \ell_{\mathrm{LLN}}(h^\star(\mathbf{x}), k)$ means $p_k(\mathbf{x}) = 0$ for $k \neq y$ and $p_y(\mathbf{x}) = 1$ for $k \neq y$. It means that the optimal classifier $\tilde{h}^\star$ from noisy data can make correct predictions on any inputs, which is consistent with the optimal classifier $h^\star$ obtained from clean data.

As for the condition $\ell_{\mathrm{LLN}}(\tilde{h}^\star(\mathbf{x}), k) \leq \ell_{\mathrm{LLN}}(h^\star(\mathbf{x}), k)$, we can get $p_k(\mathbf{x}) = 1$ for a $k \neq y$, which means that the optimal classifier $\tilde{h}^\star$ from noisy data cannot make correct predictions on samples $\mathbf{x} \in \mathbf{R}$. To verify this, we use the robust loss function RCE $\ell_{\mathrm{RCE}}$ as an example, and it can be easily generalized to other robust los functions mentioned above. Based on the definition of the RCE loss (Wang et al., 2019), we have

$$\ell_{\mathrm{RCE}}(\tilde{h}^\star(\mathbf{x}), k) = C_{\mathrm{RCE}}(1 - p_k(\mathbf{x}))$$
$$\ell_{\mathrm{RCE}}(h^\star(\mathbf{x}), k) = C_{\mathrm{RCE}},$$

where $C_{\mathrm{RCE}} > 0$ is a constant. The above equations show that any $0 \leq p_k(\mathbf{x}) \leq 1$ can make the condition $\ell_{\mathrm{LLN}}(\tilde{h}^\star(\mathbf{x}), k) \leq \ell_{\mathrm{LLN}}(h^\star(\mathbf{x}), k)$ hold. Meanwhile, $\tilde{h}^\star$ is the global minimizer of the risk over the noisy data, which makes $\tilde{h}^\star$ memorize the noisy dataset.

Therefore, $\tilde{h}^\star$ makes incorrect predictions for $\mathbf{x} \in \mathbf{R}$ such that $p_k(\mathbf{x}) = 1$ for a $k \neq y$, and $h^\star$ is the global optimal over clean data, which gives correct predictions for $\mathbf{x} \in \mathbf{R}$ such that $p_k(\mathbf{x}) = 1$ for a $k = y$. That completes the proof as $h^\star$ makes different predictions on $\mathbf{x} \in \mathbf{R}$ compared to $\tilde{h}^\star$. $\qquad \square$

## C  PROOFS FOR THEOREM 4.2

*Proof.* **To begin with, we show the first part.** Let samples $\mathbf{x}_i = y_i(\boldsymbol{\mu} - \sigma \mathbf{z}_i)$, where $\mathbf{z} \sim \mathcal{N}(0, \mathbf{I}_d)$. Then we will show that $-\boldsymbol{\mu}^\top \nabla_\theta \mathcal{L}(\theta_t)$ is lower bounded by a positive number.

Since $\mathbf{x}_i$ is sampled from standard normal distribution, $\frac{1}{n} \sum_{i=1}^n \tilde{y}_i \boldsymbol{\mu}^\top \mathbf{x}_i$ has limited variance. By the law of large number, $\frac{1}{n} \sum_{i=1}^n \tilde{y}_i \boldsymbol{\mu}^\top \mathbf{x}_i$ converges in probability to its mean.

Note that $\mathbf{x}|y = 1$ is a Gaussian random vector with independent entries, we have $\mathbf{x}^\top \boldsymbol{\mu} \overset{\mathrm{d}}{=} w + 1$, where $w \sim \mathcal{N}(0, \sigma^2)$. Note that $r < 1$, which means that most half of samples are mislabeled.

Now we deal with the in Eq. (**??**).

$$\frac{1}{2n}|\boldsymbol{\mu}^\top\big(\sum_{i=1}^n \tanh(\theta_t^\top x_i)\big)| = \frac{1}{2n}|\mathbf{q}^\top\mathbf{p}|$$
$$\leq \frac{1}{2n}\mathbf{q}\mathbf{p}, \tag{25}$$

$\mathbf{q} = (\boldsymbol{\mu}^\top\mathbf{x}_1, \boldsymbol{\mu}^\top\mathbf{x}_2, \dots, \boldsymbol{\mu}^\top\mathbf{x}_n) \in \mathbb{R}^n$, and $\mathbf{p} = (\tanh(\theta_t^\top x_1), \tanh(\theta_t^\top x_2), \dots, \tanh(\theta_t^\top x_n)) \in \mathbb{R}^n$.

By triangle inequality of the norm,

$$\mathbf{q} = \mathbf{q} - \mathbf{1} + \mathbf{1} \leq \mathbf{q} - \mathbf{1} + \mathbf{1} = \sqrt{n} + \mathbf{q} - \mathbf{1},$$

where $\mathbf{q} - \mathbf{1}$ is a random vector with Gaussian coordinates. By Lemma **??**,

$$\mathbf{q} - \mathbf{1}/\sigma \leq 2\sigma\sqrt{n} \tag{26}$$

with probability $1 - \delta$ when $n \geq c_1 \log 1/\delta$, where $c_1$ is a constant.

On the other hand,

$$\mathbf{p} - \tanh(\theta_t^\top\boldsymbol{\mu})\mathbf{1}_n + \tanh(\theta_t^\top\boldsymbol{\mu})\mathbf{1}_n \leq \tanh(\theta_t^\top\boldsymbol{\mu})\mathbf{1}_n + \mathbf{p} - \tanh(\theta_t^\top\boldsymbol{\mu})\mathbf{1}_n$$
$$\leq \tanh(\theta_t^\top\boldsymbol{\mu})\mathbf{1}_n + \theta_t\mathbf{q} - \mathbf{1}$$
$$= \tanh(\theta_t^\top\boldsymbol{\mu})\sqrt{n} + 2\sigma\sqrt{n}\theta_t, \tag{27}$$

Then we take Eq. (25) and Eq.(27) together, we can obtain:

$$-\nabla_\theta\mathcal{L}(\theta_t)^\top\boldsymbol{\mu} \geq \frac{1}{2}\mathrm{Erf}[\frac{1-r}{\sqrt{2}\sigma}] + \frac{\sigma}{\sqrt{2\pi}}\exp\big(-\frac{(r-1)^2}{2\sigma^2}\big) - \sigma(\tanh(\theta_t^\top\boldsymbol{\mu}) + 2\sigma\theta_t) \tag{28}$$

By Lemma 8 from Liu et al. (2020), we have $\sup_{\theta\in\mathbb{R}^d}\nabla_\theta\mathcal{L}(\theta) \leq 1 + 2\sigma$. Therefore, Eq. (28) can be rewritten as:

$$\frac{-\nabla_\theta\mathcal{L}(\theta_t)^\top\boldsymbol{\mu}}{\nabla_\theta\mathcal{L}(\theta_t)} \geq \frac{\mathrm{Erf}[\frac{1-r}{\sqrt{2}\sigma}] + 2\frac{\sigma}{\sqrt{2\pi}}\exp\big(-\frac{(r-1)^2}{2\sigma^2}\big)}{1 + 2\sigma} - \frac{\sigma(\tanh(\theta_t^\top\boldsymbol{\mu}) + 2\sigma\theta_t)}{1 + 2\sigma}$$
$$\geq \frac{b_0}{1 + 2\sigma} - \frac{\sigma(\tanh(\theta_t^\top\boldsymbol{\mu}) + 2\sigma\theta_t)}{1 + 2\sigma}, \tag{29}$$

where we let $b_0 = \frac{1}{2}\mathrm{Erf}[\frac{1-r}{\sqrt{2}\sigma}] + \frac{\sigma}{\sqrt{2\pi}}\exp\big(-\frac{(r-1)^2}{2\sigma^2}\big)$.

Then we prove $\frac{-\nabla_\theta\mathcal{L}(\theta_t)^\top\boldsymbol{\mu}}{\nabla_\theta\mathcal{L}(\theta_t)} \geq \frac{1}{10}\frac{b_0}{1+2\sigma}$ by mathematical induction, which can help us get rid of the dependence on $\theta_t$ for the lower bound in Eq. (29).

For $t = 0$, the inequality holds trivially. By the gradient descent algorithm, $\theta_{t+1} = -\eta\sum_{i=0}^t\nabla_\theta\mathcal{L}(\theta_i)$, where $-\boldsymbol{\mu}^\top\nabla_\theta\mathcal{L}(\theta_i)/\nabla_\theta\mathcal{L}(\theta_i) \geq \frac{1}{10}\frac{b_0}{1+2\sigma}$.

$$\frac{\theta_{t+1}^\top\boldsymbol{\mu}}{\theta_{t+1}} \geq \frac{-\eta\sum_{i=0}^t\boldsymbol{\mu}^\top\nabla_\theta\mathcal{L}(\theta_i)}{\eta\sum_{i=0}^t\nabla_\theta\mathcal{L}(\theta_i)}$$
$$\geq \frac{\frac{1}{10}\frac{b_0}{1+2\sigma}(\sum_{i=0}^t\nabla_\theta\mathcal{L}(\theta_i))}{\sum_{i=0}^t\nabla_\theta\mathcal{L}(\theta_i)}$$
$$\geq \frac{1}{10}\frac{b_0}{1+2\sigma}$$

As $t + 1 < T$, we have $\theta_{t+1} \le 10 \frac{1+2\sigma}{b_0} \theta_{t+1}^\top \boldsymbol{\mu} \le \frac{1+2\sigma}{b_0}$. Taking it into Eq. (29), we have

$$\frac{-\nabla_\theta \mathcal{L}(\theta_t)^\top \boldsymbol{\mu}}{\nabla_\theta \mathcal{L}(\theta_t)} \ge \frac{b_0}{1+2\sigma} - \frac{\sigma(0.1 + \frac{1+2\sigma}{b_0})}{1+2\sigma}$$

To show $\frac{-\nabla_\theta \mathcal{L}(\theta_t)^\top \boldsymbol{\mu}}{\nabla_\theta \mathcal{L}(\theta_t)}$ is lower bounded by $\frac{1}{10} \frac{b_0}{1+2\sigma}$, we need to have

$$h(\sigma) = \frac{9}{10} \frac{b_0}{1+2\sigma} - \sigma(0.1 + \frac{1+2\sigma}{b_0}) > 0$$

It is straightforward to verify that $h(\sigma = 0) > 0$ and it can be verified that when $0 < \sigma < c_0$, we have $h'(\sigma) > 0$. Therefore, for $0 < \sigma < c_0$ and any $t < T - 1$

$$\frac{-\nabla_\theta \mathcal{L}(\theta_t)^\top \boldsymbol{\mu}}{\nabla_\theta \mathcal{L}(\theta_t)} \ge \frac{1}{10} \frac{b_0}{1+2\sigma}$$

Hence by gradient descent algorithm $\theta_T = -\eta \sum_{i=0}^{T-1} \nabla_\theta \mathcal{L}(\theta_i)$ and the same proof above, we have

$$\frac{\theta_T^\top \boldsymbol{\mu}}{\theta_T} \ge \frac{1}{10} \frac{b_0}{1+2\sigma} \tag{30}$$

**For the second part:** the prediction accuracy on mislabeled sample set $B$ converges in probability to its mean. Therefore, the expectation of the prediction accuracy on mislabeled samples is given by

$$\begin{aligned}
[\mathbb{1}\{\text{sign}(\theta_T^\top \mathbf{x}) = y\}] &= [\mathbb{1}\{\text{sign}(y\theta_T^\top(\boldsymbol{\mu} - \sigma\mathbf{z})) = y\}] \\
&= [\mathbb{1}\{\text{sign}(\theta_T^\top(\boldsymbol{\mu} - \sigma\mathbf{z})) = 1\}] \\
&= \Pr[\sigma\theta_T^\top \mathbf{z} > \theta_T^\top \boldsymbol{\mu}]
\end{aligned} \tag{31}$$

Note that $\mathbf{z}$ is a standard Gaussian vector, $\theta_T^\top \mathbf{z}$ is distributed as $\mathcal{N}(0, \theta_T{}^2)$ Thus, Eq. (31) is equivalent to $\Phi(\frac{\theta_T^\top \boldsymbol{\mu}}{\sigma\theta_T})$.

By the inequality $1 - \Phi(x) \le \exp\{-x^2/2\}$ for $x > 0$, then we have

$$\Phi(\frac{\theta_T^\top \boldsymbol{\mu}}{\sigma\theta_T}) \ge 1 - \exp\{-\frac{(\frac{\theta_T^\top \boldsymbol{\mu}}{\sigma\theta_T})^2}{2}\} \ge 1 - \exp\{-\frac{1}{200}(\frac{b_0}{(1+2\sigma)\sigma})^2\}$$

We denote $g(\sigma)$ by:

$$g(\sigma) = \frac{\text{Erf}[\frac{1-r}{\sqrt{2}\sigma}]}{2(1+2\sigma)\sigma} + \frac{\exp(-\frac{(r-1)^2}{2\sigma^2})}{\sqrt{2\pi}(1+2\sigma)},$$

where $g(\sigma) > 0$ for any $\sigma > 0$. Note that $g(\sigma) \to \infty$ when $\sigma \to 0$, and $g(\sigma)$ is monotone decreasing as $\sigma$ increases since $g'(\sigma) < 0$ for $\sigma > 0$.

$\square$

## D PROOFS FOR THEOREM 5.1

*Proof.* The cross-entropy between the conditional distributions $p$ and $q_h$ is given by

$$\mathbb{E}_{\mathbf{z} \sim p(\mathbf{z})} [H(p(\cdot|\mathbf{z}), q_h(\cdot|\mathbf{z}))] \tag{32}$$

$$= \mathbb{E}_{\mathbf{z} \sim p(\mathbf{z})} \left[ \mathbb{E}_{\tilde{\mathbf{z}} \sim p(\tilde{\mathbf{z}}|\mathbf{z})} [-\log q_h(\tilde{\mathbf{z}}|\mathbf{z})] \right] \tag{33}$$

$$= \mathbb{E}_{\tilde{\mathbf{z}}, \mathbf{z} \sim p(\tilde{\mathbf{z}}, \mathbf{z})} \left[ -\frac{1}{\tau} h(\tilde{\mathbf{z}})^\top h(\mathbf{z}) + \log C_h(\mathbf{z}) \right] \tag{34}$$

$$= -\frac{1}{\tau} \mathbb{E}_{\tilde{\mathbf{z}}, \mathbf{z} \sim p(\tilde{\mathbf{z}}, \mathbf{z})} \left[ h(\tilde{\mathbf{z}})^\top h(\mathbf{z}) \right] + \mathbb{E}_{\mathbf{z} \sim p(\mathbf{z})} [\log C_h(\mathbf{z})]. \tag{35}$$

Using the definition of $C_h$, we obtain

$$= -\frac{1}{\tau} \mathop{\mathbb{E}}_{\tilde{\mathbf{z}}, \mathbf{z} \sim p(\tilde{\mathbf{z}}, \mathbf{z})} \left[ h(\tilde{\mathbf{z}})^\top h(\mathbf{z}) \right] \tag{36}$$

$$+ \mathop{\mathbb{E}}_{\mathbf{z} \sim p(\mathbf{z})} \left[ \log \int_{\mathcal{Z}} e^{h(\tilde{\mathbf{z}})^\top h(\mathbf{z})/\tau} \right]. \tag{37}$$

By assumption the marginal distribution is uniform, i.e., $p(\mathbf{z}) = |\mathcal{Z}|^{-1}$. We expand by $|\mathcal{Z}||\mathcal{Z}|^{-1}$ and estimate the integral by sampling from $p(\mathbf{z}) = |\mathcal{Z}|^{-1}$, yielding

$$= -\frac{1}{\tau} \mathop{\mathbb{E}}_{\tilde{\mathbf{z}}, \mathbf{z} \sim p(\tilde{\mathbf{z}}, \mathbf{z})} \left[ h(\tilde{\mathbf{z}})^\top h(\mathbf{z}) \right] \tag{38}$$

$$+ \mathop{\mathbb{E}}_{\mathbf{z} \sim p(\mathbf{z})} \left[ \log |\mathcal{Z}| \mathop{\mathbb{E}}_{\tilde{\mathbf{z}} \sim p(\tilde{\mathbf{z}})} \left[ e^{h(\tilde{\mathbf{z}})^\top h(\mathbf{z})/\tau} \right] \right] \tag{39}$$

$$= -\frac{1}{\tau} \mathop{\mathbb{E}}_{\tilde{\mathbf{z}}, \mathbf{z} \sim p(\tilde{\mathbf{z}}, \mathbf{z})} \left[ h(\tilde{\mathbf{z}})^\top h(\mathbf{z}) \right] \tag{40}$$

$$+ \mathop{\mathbb{E}}_{\mathbf{z} \sim p(\mathbf{z})} \left[ \log \mathop{\mathbb{E}}_{\tilde{\mathbf{z}} \sim p(\tilde{\mathbf{z}})} \left[ e^{h(\tilde{\mathbf{z}})^\top h(\mathbf{z})/\tau} \right] \right] + \log |\mathcal{Z}|. \tag{41}$$

By inserting the definition $h = f \circ g$,

$$= -\frac{1}{\tau} \mathop{\mathbb{E}}_{\tilde{\mathbf{z}}, \mathbf{z} \sim p(\tilde{\mathbf{z}}, \mathbf{z})} \left[ (f \circ g)(\tilde{\mathbf{z}})^\top (f \circ g)(\mathbf{z}) \right] \tag{42}$$

$$+ \mathop{\mathbb{E}}_{\mathbf{z} \sim p(\mathbf{z})} \left[ \log \mathop{\mathbb{E}}_{\tilde{\mathbf{z}} \sim p(\tilde{\mathbf{z}})} \left[ e^{(f \circ g)(\tilde{\mathbf{z}})^\top (f \circ g)(\mathbf{z})/\tau} \right] \right] \tag{43}$$

$$+ \log |\mathcal{Z}|, \tag{44}$$

$$\tag{45}$$

By assumption, $q_{\mathrm{h}}(\tilde{\mathbf{z}}|\mathbf{z})$ is powerful enough to match $p(\tilde{\mathbf{z}}|\mathbf{z})$ for the correct choice of $h$ — in particular, for $h(\mathbf{z}) = \sqrt{\tau \kappa} \mathbf{z}$. The global minimum of the cross-entropy between two distributions is reached if they match by value and have the same support. Thus, this means

$$p(\tilde{\mathbf{z}}|\mathbf{z}) = q_{\mathrm{h}}(\tilde{\mathbf{z}}|\mathbf{z}). \tag{46}$$

This expression also holds true for $\tilde{\mathbf{z}} = \mathbf{z}$; additionally using that $h$ maps from a unit hypersphere to one with radius $\sqrt{\tau \kappa}$ yields

$$p(\mathbf{z}|\mathbf{z}) = q_{\mathrm{h}}(\mathbf{z}|\mathbf{z}) \tag{47}$$

$$\Leftrightarrow \quad C_p^{-1} e^{\kappa \mathbf{z}^\top \mathbf{z}} = C_h(\mathbf{z})^{-1} e^{h(\mathbf{z})^\top h(\mathbf{z})/\tau} \tag{48}$$

$$\Leftrightarrow \quad C_p^{-1} e^\kappa = C_h(\mathbf{z})^{-1} e^\kappa \tag{49}$$

$$\Leftrightarrow \quad C_p = C_h. \tag{50}$$

As the normalization constants are identical we get for all $\mathbf{z}, \tilde{\mathbf{z}} \in \mathcal{Z}$

$$e^{\kappa \mathbf{z}^\top \tilde{\mathbf{z}}} = e^{h(\mathbf{z})^\top h(\tilde{\mathbf{z}})} \Leftrightarrow \kappa \mathbf{z}^\top \tilde{\mathbf{z}} = h(\mathbf{z})^\top h(\tilde{\mathbf{z}}). \tag{51}$$

First, we begin with the case $r = 1$. As $h$ maintains the dot product we have:

$$\forall \mathbf{z}, \tilde{\mathbf{z}} \in \mathcal{Z} : \mathbf{z}^\top \tilde{\mathbf{z}} = h(\mathbf{z})^\top h(\tilde{\mathbf{z}}). \tag{52}$$

We consider the partial derivative w.r.t. $\mathbf{z}$ and obtain:

$$\forall \mathbf{z}, \tilde{\mathbf{z}} \in \mathcal{Z} : \tilde{\mathbf{z}} = \mathbf{J}_h^\top(\mathbf{z}) h(\tilde{\mathbf{z}}). \tag{53}$$

Taking the partial derivative w.r.t. $\tilde{\mathbf{z}}$ yields

$$\forall \mathbf{z}, \tilde{\mathbf{z}} \in \mathcal{Z} : \mathbf{I} = \mathbf{J}_h^\top(\mathbf{z}) \mathbf{J}_h(\tilde{\mathbf{z}}). \tag{54}$$

We can now conclude

$$\forall \mathbf{z}, \tilde{\mathbf{z}} \in \mathcal{Z} : \mathbf{J}_h(\tilde{\mathbf{z}})^{-1} = \mathbf{J}_h^\top(\mathbf{z}). \tag{55}$$

which implies a constant Jacobian matrix $\mathbf{J}_h(\mathbf{z}) = \mathbf{J}_h$ as the identity holds on all points in $\mathcal{Z}$, and further that the Jacobian $\mathbf{J}_h$ is orthogonal. Hence, $\forall \mathbf{z} \in \mathcal{Z} : h(\mathbf{z}) = \mathbf{J}_h \mathbf{z}$ is an orthogonal linear transformation.

Finally, for $r \neq 1$ we can leverage the previous result by introducing $h'(\mathbf{z}) := h(\mathbf{z})/r$. For $h'$ the previous argument holds, implying that $h'$ is an orthogonal transformation. Therefore, the restriction of $h$ to $\mathcal{Z}$ is an orthogonal linear transformation scaled by $r^2$.

Thus, $f$ recovers the latent sources up to orthogonal linear transformations, concluding the proof.

$\square$

