# OpenReview forum: "When Self-Supervised Learning Meets Unbounded Pseudo-Label Generation"
_ICLR.cc/2024/Conference — ICLR 2024 Conference Withdrawn Submission_

### Official Review · Reviewer_bjmg · 2023-10-25

**Soundness:** 3 good
**Presentation:** 2 fair
**Contribution:** 2 fair
**Rating:** 5
**Confidence:** 2

**Summary:**

This paper studies instance discrimination-based self-supervised learning, where several augmented views are pulled together and negative samples are pushed away from each other. One of the key challenges is the difficulty in determining positive and negative samples precisely. To this end, the authors propose to assign pseudo-labels for each instance and prove that those pseudo-labels are unbounded. Under such a setting, the authors further come up with a robust learning paradigm that manages to model the dynamic relationship precisely. Empirical evaluation on several self-supervised benchmarks demonstrate the effectiveness of the proposed method.

**Strengths:**

1. The motivation is clear.
2. The structure of this paper is well organized.
3. A detailed theoretical analysis is provided.
4. The improvements of the proposed method over baselines are somewhat significant.

**Weaknesses:**

I am not an expert on theoretical analysis. Therefore, please pay more attention to addressing concerns from other reviewers. Here are my concerns:

1. Referring to instance discrimination (ID) as SSL is somewhat an overstatement. I recognize that the scope of this paper is the theoretical analysis of ID-based SSL methods, but there also exists a wide range of other SSL methods such as masked autoencoding [A, B], predictive learning [C, D], and image colorizing [E]. It is better to replace SSL with ID.

2. This paper contains some overclaims. The authors claim that the proposed method manages to "model the dynamic relationship *precisely*". Similar claims appear frequently in this paper. However, it is almost impossible without ground-truths.


References

[A] K. He et al. Masked autoencoders are scalable vision learners. In CVPR 2022.

[B] H. Wang et al. Hard patches mining for masked image modeling. In CVPR 2023.

[C] M. Noroozi and P. Favaro. Unsupervised learning of visual representations by solving jigsaw puzzles. In ECCV 2016.

[D] H. Wang et al. DropPos: Pre-Training Vision Transformers by Reconstructing Dropped Positions. In NeurIPS 2023.

[E] R. Zhang et al. Colorful image colorization. In ECCV 2016.

**Questions:**

I do not have further questions. Please refer to the weaknesses section.

---

### Official Review · Reviewer_fBvz · 2023-11-02

**Soundness:** 3 good
**Presentation:** 2 fair
**Contribution:** 2 fair
**Rating:** 3
**Confidence:** 4

**Summary:**

The paper focuses on capturing the relationships between instances. Specifically, the authors propose to generate pseudo-labels to determine whether two augmented instances belong to the same category. The pseudo-labels are proved to be unbounded, but the negative effect can be ignored if the variances are significantly small. On top of this, the paper proposes to optimize an SSL model in an iterative update manner.

**Strengths:**

1. Different from the methods that use clustering to capture the intra-sample relationships, this work proposes a novel contrastive-based method. It avoids tuning the clustering hyperparameters;
2. The provided theoretical results explain the role of the noisy pseudo labels;
3. The proposed method shows consistent improvements in various settings.

**Weaknesses:**

1. The main concern is the formulation of the objective function Eq. (7) and the corresponding update rules: The constraint requires $f, f_{ph}$ to minimize $\mathcal{L}\_{PAR}$, which sets $f, f\_{ph}$ to the global minimizer, while the outer objective requires them to be minimizer of $\mathcal{L}\_{align} + \mathcal{L}\_{prior}$. It is not a bi-level optimization problem but a combination of two problems. As a result, it is hard to determine whether the update rules are reasonable.
2. At the beginning of Sec. 4.3, it is claimed that the mislabeling rate can approach 1, however, according to Theorem 4.1 we can only conclude that the misleading rate is greater than 0.
3. In Theorem 4.2, $g(\sigma) \rightarrow \infty$ as $\sigma \rightarrow 0$ is not obvious since both the numerator and denominator tend to 0. Please provide more details.
4. The novelty of the proposed Precise Adjustment Regularization (PAR) needs more explanation. It seems that $\mathcal{L}_{PAR}$ directly uses the discretized similarity as pseudo labels. Besides, the motivation for using moving average predictions as pseudo labels seems unclear.
5. In Sec. 6.2, the objective $\mathcal{L}\_{SSL} + \mathcal{L}\_{PAR}$ is shown to be less effective than the iterative update. However, the computational cost of the iterative update is double according to my understanding. Therefore, it should be compared under the same computational cost for a fair comparison.

**Questions:**

Please refer to the weaknesses.

---

### Official Review · Reviewer_ZMGw · 2023-11-09

**Soundness:** 2 fair
**Presentation:** 2 fair
**Contribution:** 2 fair
**Rating:** 5
**Confidence:** 4

**Summary:**

One of the challenges in SSL is the lack of labeled information, which can impact SSL performance and result in inconsistencies in the label space. For instance, SSL methods tend to focus on combining augmented samples from the same ancestral source, while ignoring the combination of augmented samples derived from ancestor samples that ***have the same labels***. To tackle this problem, the authors proposed a new SSL mechanism that generates pseudo-labels to guide learning, thereby reducing label space inconsistencies and improving the robustness of SSL with pseudo-labels' noise at the early stage of learning. The authors also integrated PAR with existing SSL methods to induce better representations and tested their framework on various tasks. The findings revealed that their approach significantly enhanced the performance of existing SSL methods.

**Strengths:**

* Originality: In most of the existing SSL methods, we treat each sample as an independent class due to the absence of labeled information. This paper proposed a novel approach to enhance SSL methods by generating pseudo-labels to guide model learning better in training. This new notion to leverage pseudo-labels reveals its creativity and originality.
* Quality: The numbers in their experiments were evaluated with diverse well-known existing SSL methods and showcased the effectiveness of the proposed framework. The proposed method improved SSL, which demonstrates the paper quality.

**Weaknesses:**

* In the abstract, the statement *"it is difficult for SSL to accurately gather samples of the same category and separate samples of different categories in the training stage"*, is causing confusion. The interpretation behind this is quite different from the challenge presented in the introduction. It would be helpful to provide more examples to clarify the challenge presented in the introduction.
* I recommend including a formal mathematical definition when introducing the concept of pseudo-label generation, specifically in the context of discussing both the target network and online network. What exact the online network is. For example, in Section 4.4, it is suggested to move the formula for generating pseudo-labels earlier in the discussion.
* Further clarification and details are required to support the PAR regarding the $out$ function introduced in Eq. 5.
* This paper can be challenging to follow because of its excessively long sentences.

**Questions:**

* What is the rate of mislabeling in the empirical experiments? This shows the effectiveness of the pseudo-label generation.